# Effects of Weak and Strong Drought Conditions on Physiological Stability of Flowering Soybean

**DOI:** 10.3390/plants11202708

**Published:** 2022-10-13

**Authors:** Shuang Song, Zhipeng Qu, Xinyu Zhou, Xiyue Wang, Shoukun Dong

**Affiliations:** College of Agriculture, Northeast Agricultural University, Harbin 150030, China

**Keywords:** soybean, antioxidant enzymes, membrane lipid peroxidation, osmotic adjustment, strong and weak drought

## Abstract

Soybean is an important food crop in the world. Drought can seriously affect the yield and quality of soybean; however, studies on extreme drought—weak and strong—are absent. In this study, drought-tolerant soybean Heinong 44 (HN44) and sensitive soybean Heinong 65 (HN65) were used as the test varieties, and the effects of strong and weak droughts on the physiological stability of soybean were explored through the drought treatment of soybean at the early flowering stage. The results showed that the contents of malondialdehyde (MDA), hydrogen peroxide (H_2_O_2_), and superoxide anions ($O_{2}^{\cdot -}$) increased with the increase in the degree of drought. The plant height and relative water content decreased, and photosynthesis was inhibited. The activities of superoxide dismutase (SOD), peroxidase (POD), and catalase (CAT), and the total antioxidant capacity (T-AOC) showed a trend of first increasing and then decreasing. Through contribution analysis, CAT changed the most, and the role of SOD gradually increased with the aggravation of drought. With the aggravation of drought, the contents of soluble sugar (SSC) and proline (Pro) increased gradually, and the content of soluble protein (SP) increased initially and then decreased. According to contribution analysis, SSC had the highest contribution to osmotic adjustment. SSC and Pro showed an upward trend with the aggravation of drought, indicating that their role in drought was gradually enhanced.

## 1. Introduction

Soybean (*Glycine max* (Linn.) Merr.) is an important grain and oil crop that plays an important role in food, pharmaceutical, feed processing, and bioenergy production, and is widely cultivated worldwide [1]. It contains 40% protein and 20% oil [2], with the dual attributes of providing protein and oil raw materials [3]. With a preference for a healthy diet, consumers are increasingly attracted to the high-quality protein offered by soybean. China’s soybean demand and yield increased simultaneously in 2020, with the annual demand reaching 120 million tons (an increase of 12.6% compared with 2019) and total domestic production reaching 1960.18 million tons (an increase of 8.4% compared with 2019). Despite the increase in the annual output, about 90% of the soybean demand is dependent on imports [4], and the reasons for this phenomenon are diverse.

In a farmland ecosystem, crops are inevitably affected by various abiotic stresses. Stress has a huge impact on plant growth and survival by affecting multiple biological processes in plants, and the effects caused by different stress times and stress intensities are also different [5]. Drought, one of the typical abiotic stresses, is the most important factor affecting the crop yield and quality [6]. Zou et al. [7] found that the yield loss of soybean caused by drought could reach 25–50%. According to the occurrence of drought and the degree of impact, with reference to “Grade of agricultural drought” [8], the water potential is an important basis for judging the level of drought. In recent years, the studies on drought resistance in soybean mainly focused on moderate drought, and there have been few systematic studies on weak drought (light drought) and strong drought (heavy). Soybean can counter a certain degree of weak drought by regulating osmotic adjustment and its antioxidant system to reduce the damage caused by the drought. Losses accrued due to strong drought are comparatively more severe than those accrued from weak drought [9]. Due to global warming and the considerable variation in precipitation laws, the degree of drought stress, the probability of a strong drought, and the drought duration have increased [10,11], posing a serious threat to food security in various countries [12].

Drought causes water imbalance and the accumulation of reactive oxygen species (ROS) in plants, leading to cell membrane damage and even plant death due to membrane lipid peroxidation. When organisms are subjected to drought stress, the generation and elimination of ROS are unbalanced. In soybean, the contents of hydrogen peroxide (H_2_O_2_) and superoxide anions ($O_{2}^{\cdot -}$) are usually used to measure the damage degree of plants by ROS [13]. Membrane lipid peroxidation and cell damage are caused when free oxygen radicals attack unsaturated fatty acids in the biofilm. Malondialdehyde (MDA), as the final product of membrane lipid peroxidation, indirectly reflects the degree of cell damage. The production of ROS is the inevitable result of aerobic metabolism in plants [14]. ROS participate in various pathways of plant growth and development as a second messenger. Under low-concentration conditions, ROS are harmless to cells [15]. However, when plants are under stress conditions, the ROS content exceeds their own defense range, and the cells will be in an oxidative stress state, resulting in protein oxidation, nucleic acid damage, and programmed cell death under severe conditions. Yang et al. [16] found that the contents of H_2_O_2_, $O_{2}^{\cdot -}$, and MDA in leaves increased with the aggravation of stress through maize drought treatment, and their contents can be used to judge the degree of leaf damage.

Plants have evolved complex physiological and biochemical systems, such as ROS scavenging systems composed of multiple enzymes, to resist drought stress. Drought leads to metabolic disorders in plants, resulting in imbalances in the production and removal of ROS in plants. Antioxidant enzymes must eliminate excessive ROS to prevent damage and plant death [17]. The antioxidant enzyme system mainly includes superoxide dismutase (SOD), catalase (CAT), and peroxidase (POD). ROS scavenging by the antioxidant enzyme system alleviates cell damage. Further, an osmotic adjustment system and drought cause excessive dehydration in plants. Rezayian et al. [18] found that proline (Pro), soluble sugar (SSC), and soluble protein (SP) can be used as organic solutes for osmotic adjustment to improve the cell concentration, maintain the osmotic balance between plants and the environment, and prevent excessive dehydration to enable plants to adapt to a certain degree of drought.

In this study, two soybean varieties with different drought resistances, drought-tolerant Heinong 44 (HN44) and drought-sensitive Heinong 65 (HN65), were selected for drought stress treatment with various degrees at the early flowering stage. The early flowering stage of the soybean marked the transformation from vegetative growth to reproductive growth, which was a crucial period for the growth and development of the soybean. In this study, the effects of weak drought and strong drought on the physiological homeostasis of soybean varieties with different drought resistances were analyzed, and the impact of varying drought degrees on membrane lipid peroxidation, the antioxidant system, and osmotic adjustment substances in soybean was determined. Furthermore, the contribution of the two soybean antioxidant enzymes and osmotic adjustment substances was analyzed to explore the contribution of the regulatory substances in the two systems, which provided a theoretical basis for soybean drought resistance cultivation and variety screening.

## 2. Results

### 2.1. Effect of Drought on Membrane Lipid Peroxidation of Soybean

The effects of weak drought and strong drought on the membrane lipid peroxidation of soybean leaves at the early flowering stage are shown in Figure 1, Figure 2 and Figure 3. In the control group, the contents of MDA, H_2_O_2_, and $O_{2}^{\cdot -}$ in soybean leaves remained unchanged, and the contents of MDA, H_2_O_2_, and $O_{2}^{\cdot -}$ increased gradually with the extension of drought time and the aggravation of the drought degree. Under severe drought conditions of Ψ = −0.6 MPa and Ψ = −0.86 MPa, soybean plants died on the fourth and seventh days, so data could not be obtained.

The results of this experiment showed no significant change in the MDA content of the two varieties throughout the treatment at Ψ = 0.00 MPa. With the prolongation of stress time and the aggravation of the stress degree, the content of MDA gradually increased. The contents of MDA in HN44 and HN65 reached the maximum value on the tenth day of stress, which were 80.58% and 90.36% higher than those in the control group, respectively, at Ψ = −0.20 MPa. The MDA content of the drought-resistant variety HN44 was lower than that of the sensitive variety HN65, indicating that HN44 was less affected by the membrane lipid peroxidation than HN65. On the fourth day of stress treatment, the MDA content of the strong drought treatment increased by 23.97% compared with that of the weak drought treatment.

At Ψ = 0.00 MPa, the H_2_O_2_ content of the two varieties did not change significantly during the whole process; however, the H_2_O_2_ content increased gradually with the extension of the stress time and the aggravation of stress. At Ψ = −0.86 MPa, the H_2_O_2_ content reached the maximum on the fourth day of stress, which was 139.13% and 154.13% higher than that in the control group. The increase in the H_2_O_2_ content in the drought-resistant HN44 was smaller than that in the drought-sensitive HN65. On the fourth day of stress treatment, the H_2_O_2_ content of the strong drought treatment increased by 110.05% compared with that of the weak drought treatment.

At Ψ = 0.00 MPa, the $O_{2}^{\cdot -}$ content of the two varieties did not change significantly during the whole process. The $O_{2}^{\cdot -}$ content increased gradually with the extension of stress time and the aggravation of stress. The content of $O_{2}^{\cdot -}$ in HN44 reached the maximum on the seventh day of stress, increasing by 867.7% compared with the control group, at Ψ = −0.60 MPa. At Ψ = −0.86 MPa, it reached the maximum on the fourth day of stress, increasing by 891.7% compared with the control group. The variation in $O_{2}^{\cdot -}$ in the drought-resistant varieties was less than that in the sensitive varieties. On the fourth day of stress treatment, the $O_{2}^{\cdot -}$ content of the strong drought treatment increased by 523.41% from that of the weak drought treatment.

### 2.2. Effects of Different Degrees of Drought on the Leaf Relative Water Content and Plant Height of Soybean

The changes in the leaf relative water content and plant height of soybean under drought stress are shown in Figure 4, Figure 5 and Figure 6. The relative water content of soybean leaves in the control group remained stable, and the plant height increased with time. With the increase in drought time and degree, the relative water content of the leaves decreased gradually. Under weak drought conditions (Ψ = −0.10 and −0.20 MPa), the relative water content of HN44 and HN65 leaves decreased by 14.32% and 15.81% on average compared with that of the control group. Under strong drought conditions (Ψ = −0.60 and −0.86 MPa), the relative water content of HN44 and HN65 leaves decreased by 78.89% and 82.71% on average compared with that of the control group. Under weak drought conditions (Ψ = −0.10 and −0.20 MPa), the plant height of HN44 and HN65 decreased by 3.81% and 4.23% on average compared with that of the control group. Under strong drought conditions (Ψ = −0.60 and −0.86 MPa), the plant height of HN44 and HN65 decreased by 17.40% and 17.02% on average compared with that of the control group.

### 2.3. Effects of Different Drought Conditions on Photosynthetic Parameters of Soybean

The changes in the relative chlorophyll content (SPAD) and non-photochemical quenching (NPQ) of soybean leaves under strong and weak drought conditions are shown in Figure 6 and Figure 7. In the control group, the SPAD and NPQ of soybean leaves remained stable. Under drought conditions, SPAD showed a downward trend and NPQ increased with the aggravation of drought. Under weak drought conditions (Ψ = −0.60 and −0.86 MPa), the SPAD in HN44 and HN65 decreased by 19.79% and 28.25% on average compared with that of the control group. Under strong drought conditions (Ψ = −0.60 and −0.86 MPa), SPAD in HN44 and HN65 decreased by 62.85% and 64.97% on average compared with that of the control group. NPQ increased with the increase in drought. Under weak drought conditions (Ψ = −0.60 and −0.86 MPa), NPQ in HN44 and HN65 increased by 52.45% and 60.34% on average compared with that of the control group. Under strong drought conditions (Ψ = −0.60 and −0.86 MPa), NPQ in HN44 and HN65 increased by 68.65% and 70.59% on average compared with that of the control group.

### 2.4. Effects of Different Drought Degrees on Antioxidant Enzyme Activities of Soybean

The effect of weak and strong drought on the antioxidant enzyme activity of soybean at the early flowering stage is shown in Figure 8, Figure 9, Figure 10 and Figure 11. In the control group, the activities of SOD, POD, and CAT in the soybean leaves remained unchanged. With the extension of drought time and the aggravation of the drought degree, the activities of SOD, POD, and CAT increased initially and then decreased, reaching the maximum value at approximately 4 days of stress. The contribution of antioxidant enzymes under drought conditions is shown in Table 1, indicating the contribution of the three antioxidant enzymes to resisting the ROS imbalance.

At Ψ = 0.00 MPa, no significant change in the SOD antioxidant activity of the two varieties was observed during the whole treatment. With the prolongation of stress time and the aggravation of the stress degree, the SOD activity first increased and then decreased. Each treatment reached the maximum SOD activity on the fourth day of stress. The SOD activity in HN44 reached the maximum at Ψ = −0.86 MPa on the fourth day of stress, which was 137.48% higher than that in the control group. The SOD activity in HN65 reached the maximum at Ψ = −0.60 MPa on the fourth day of stress, which was 142.22% higher than that in the control group. In general, the change in the SOD activity of drought-resistant HN44 was smaller than that of sensitive HN65, indicating that HN44 was less affected by drought than HN65. On the fourth day of stress treatment, the SOD content of the strong drought treatment increased by 52.26% from that of the weak drought treatment.

At Ψ = 0.00 MPa, there was no significant change in the POD antioxidant activity of the two varieties during the whole treatment. With the prolongation of stress time and the aggravation of the stress degree, the POD activity first increased and then decreased. The POD activity in HN44 reached the maximum value at Ψ = −0.60 MPa on the third day of stress, which was 170.63% higher than that in the control group. Further, HN65 reached the maximum value at Ψ = −0.60 MPa on the fifth day of stress, which was 158% higher than that in the control group. In general, the variation in the POD activity of the drought-resistant variety HN44 was greater than that of the sensitive variety HN65. On the fourth day of stress treatment, the POD content under strong drought treatment increased by 9.40% compared with that under weak drought treatment.

At Ψ = 0.00 MPa, there was no significant change in the CAT antioxidant activity of the two varieties during the whole treatment. With the prolongation of stress time and the aggravation of the stress degree, the CAT activity first increased and then decreased. The CAT activity in HN44 reached the maximum value at Ψ = −0.86 MPa on the fourth day of stress, which was 313.69% higher than that in the control group. Similarly, the CAT activity in HN65 reached the maximum value at Ψ = −0.60 MPa on the fourth day of stress, which was 656.25% higher than that in the control group. In general, the CAT activity of the drought-resistant variety HN44 was lower than that of the sensitive variety HN65. On the fourth day of stress treatment, the CAT content under strong drought treatment increased by 36.12% from that under weak drought treatment.

At Ψ = 0.00 MPa, the total antioxidant activities of the two varieties did not change significantly during the whole treatment. With the prolonged stress time and the aggravation of the stress degree, the total antioxidant activity increased first and then decreased, but the time at which the highest activity was reached appeared later. The T-AOC capacities in HN44 and HN65 at Ψ = −0.60 MPa reached the maximum value on the sixth day of stress, which were 122.11% and 151.90% higher than that in the control group, respectively. The CAT activity change in the drought-resistant variety HN44 was smaller than that of the sensitive variety HN65. On the fourth day of stress treatment, the T-AOC content of the strong drought treatment increased by 50.55% compared with that of the weak drought treatment.

The contribution of different antioxidant enzymes to drought regulation is shown in Table 1. In response to drought, the contribution of CAT was significantly higher than those of SOD and POD, indicating that CAT played an essential regulatory role in the soybean antioxidant system. At the same time, the contribution of CAT in HN65 was significantly lower under strong drought conditions than that under weak drought conditions, and there was no significant change in HN44. Therefore, we speculate that the significantly reduced contribution of CAT under strong drought conditions resulted in the lower drought resistance of HN65 than that of HN44. The contributions of SOD and POD changed differently under weak drought and strong drought. The contribution of SOD in HN44 under strong drought was significantly higher than that under weak drought, indicating that, with the aggravation of drought, the role of SOD in resisting ROS imbalances was gradually enhanced. The contribution of SOD in HN65 reached the threshold when the drought was −0.60 MPa; the contribution of SOD when this drought degree was exceeded showed a decreasing trend and the antioxidant effect decreased gradually. In HN44, the contribution of POD to the antioxidant system gradually decreased with the aggravation of drought, indicating that the role of POD gradually decreased in the process of weak drought becoming strong drought. In HN65, the contribution of POD under different drought conditions was not significantly different. Due to the different contributions of the three enzymes to different varieties and drought degrees, there were certain differences in the growth status of the two soybean varieties under drought conditions.

### 2.5. Effect of Drought on Osmotic Adjustment Substances in Soybean

The effect of weak drought and strong drought on the osmoregulation substance content of soybean at the early flowering stage is shown in Figure 12, Figure 13 and Figure 14. In the control group, the contents of SSC, SP, and Pro in soybean leaves remained unchanged. However, with the extension of drought time and the aggravation of the drought degree, the contents of SSC and Pro gradually increased, reaching the maximum level under severe drought conditions. The content of SP increased initially and then decreased with the extension of drought time and the aggravation of drought degree. The contribution of osmoregulation substances under drought conditions is shown in Table 2, indicating the contribution of three osmoregulation substances to resisting plant dehydration.

The results of this experiment showed that there was no significant change in the Pro contents of the two varieties during the whole treatment at Ψ = 0.00 MPa. With the prolongation of stress time and the aggravation of stress, the Pro content gradually increased. The Pro contents in HN44 and HN65 at Ψ = −0.20 MPa reached the maximum values on the tenth day of stress, which were 167.47% and 235.71% higher than that in the control group, respectively. Overall, the change in the Pro content in the drought-resistant variety HN44 was lower than that in the sensitive variety HN65. On the fourth day of stress treatment, the Pro content of strong drought treatment increased by 34.11% compared with that of weak drought treatment.

At Ψ = 0.00 MPa, the SSC contents of the two varieties did not change significantly during the whole treatment. However, with the prolongation of stress time and the aggravation of the stress degree, the content of SSC gradually increased, and the content of SSC in HN44 at Ψ = −0.60 MPa reached the maximum value on the seventh day of stress, which was 229.16% higher than that in the control group. The SSC content in HN65 reached the maximum value on the tenth day at Ψ = −0.20 MPa, which increased by 218.43% compared with that in the control group. Overall, there was no significant difference in the Pro content between the drought-resistant variety HN44 and sensitive variety HN65. However, the SSC content under strong drought treatment was 4.62% lower than that under weak drought treatment.

At Ψ = 0.00 MPa, the SP contents of the two varieties did not change significantly during the whole treatment process. With the prolongation of stress time and the aggravation of the stress degree, the content of SP first increased and then decreased. In general, the SP content in each treatment reached the maximum level on the fourth day of stress. The SP content in HN44 and HN65 on the fourth day reached the maximum and increased by 64.3% and 57.53% compared with that of the control group at Ψ = −0.10 MPa. The SP content of the drought-resistant variety HN44 was higher than that of the sensitive variety HN65. The SP content under the strong drought treatment increased by 42.25% compared with that under the weak drought treatment.

The contribution of different osmotic adjustment substances to drought is shown in Table 2. The contribution of SSC was significantly higher than that of Pro and SP, indicating that SSC played a major regulatory role in the soybean osmotic adjustment system. At the same time, the contributions of SSC under strong drought and weak drought were significantly different. The contribution under strong drought was significantly higher than that under weak drought, indicating that, with the aggravation of drought, the role of SSC in osmotic regulation was gradually enhanced, and the contribution of SSC in the two varieties reached the maximum at −0.60 MPa at the same time. Furthermore, the role of SSC in osmotic regulation reached the threshold, and the contribution of SSC gradually decreased when the drought degree was exceeded. The contribution of Pro under strong drought was higher than that under weak drought, indicating that the role of Pro was gradually enhanced with the aggravation of drought. The contribution of SP showed a decreasing trend, and the contribution increased when the drought was −0.86 MPa, but the change was not obvious, indicating that the role of SP in osmotic adjustment gradually weakened with the aggravation of drought. Compared with the two varieties, the contributions of SSC and SP in HN44 were higher than those in HN65, and the contribution of Pro was lower than that in HN65, which may be related to the different drought resistances of the two varieties.

### 2.6. Drought Stress on Plant Damage and Recovery Process

Figure 15 shows the damage and recovery process of plants under drought stress. The relationship between membrane lipid peroxidation, the antioxidant system, and osmotic regulation in plants is briefly summarized. Under drought stress, the ROS content in plants increased. SOD, as the first line of defense against the imbalance of ROS, was affected by the concentration of $O_{2}^{\cdot -}$ within a certain range. Under drought conditions, the concentration of $O_{2}^{\cdot -}$ increased, thereby improving the activity of SOD and rapidly diverging $O_{2}^{\cdot -}$ into H_2_O_2_ and $O_{2}^{\cdot -}$. H_2_O_2_ was transformed into H_2_O and $O_{2}^{\cdot -}$, which are harmless to plants under the action of POD and CAT. An increase in ROS leads to the aggravation of membrane lipid peroxidation in plants, the accumulation of MDA in plants, and damage to plant proteins and nucleic acids. At the same time, drought causes plant dehydration, resulting in plant height reduction, leaf wilting, and stem bending phenomena. Plants can reduce the osmotic potential of cells by accumulating Pro, SSC, and SP to prevent cell dehydration and deal with the damage caused by drought, to a certain extent.

## 3. Discussion

Drought, an important abiotic stress [19], can lead to the accumulation of ROS in plants, resulting in increased membrane lipid peroxidation and affecting the normal physiological activities of plants. Huseynova et al. [20] found that ROS could react with deoxyribonucleic acid, lipids, and proteins, leading to oxidative damage and enzyme inactivation, affecting the normal functioning of plants and increasing the probability of plant death under severe drought. In this experiment, the contents of H_2_O_2_ and $O_{2}^{\cdot -}$ showed an increasing trend under two different degrees of drought, and the increases under strong drought were significantly higher than those under weak drought, with a value of 56.60%. $O_{2}^{\cdot -}$ is an ionic ROS that can be used as the precursor of various ROS due to its instability and strong redox potential. Zhanassova et al. [21] applied drought stress to barley (*Hordeum vulgare*) and showed that the production of H_2_O_2_ and $O_{2}^{\cdot -}$ increased with the increase in stress, which was similar to the results presented in the current study. MDA can indicate the degree of membrane damage. In the control group of this experiment, the MDA content in the two soybean varieties remained basically unchanged. With the aggravation of drought, the MDA content gradually increased, and the change in the MDA content in drought-resistant varieties was smaller than that in the sensitive varieties, indicating a difference in the degree of membrane damage between the two soybean types. Wang et al. [22] found that the MDA content gradually increased with the increase in drought intensity through drought treatment of soybean and that there were certain differences in the MDA content changes in soybeans with different drought tolerances. The results were consistent with the experimental results in this paper. The results of this experiment showed that drought affected the morphology and photosynthesis of soybean, with that plant height and relative water content decreasing and inhibition of photosynthesis. Zhang et al. [23] treated soybean with drought stress, and the results were consistent with those of this experiment.

During the normal growth of plants, the ROS and antioxidant enzyme system can reach a dynamic balance [24]. When plants encounter weak drought, there will be a slight imbalance in ROS in plants. within a specific range of regulation, excessive ROS can be eliminated by increasing the activity of antioxidant enzymes. When a strong drought persists for a certain period, the production of ROS and the ability to scavenge antioxidant enzymes are unbalanced, the growth of plants is limited, and the leaves curl, wilt, or even die. In this experiment, the contents of the antioxidant enzymes SOD, POD, and CAT increased first and then decreased with the increase in the drought degree and time. During the first four days of drought, the activities of antioxidant enzymes in plants increased continuously, indicating that plants themselves had a certain ability to resist drought. When exceeding the threshold, the scavenging abilities of antioxidant enzymes and ROS were constantly unbalanced, proving the bottom-line level of antioxidant enzymes against ROS changes. Guo et al. [25] treated *Lycium ruthenicum* seedlings under drought stress, and the results showed that SOD, POD, and CAT increased first and then decreased with the extension of drought time, consistent with the experimental results in this paper. In addition, it was found that the contents of antioxidant enzymes in rape [26] and pea [27] followed similar trends.

Drought can lead to water loss in plants. Soybean can maintain its cell swelling pressure by accumulating osmotic adjustment substances, such as Pro, SS, and SP, which is beneficial to other physiological processes and plays an important role in maintaining cell growth and membrane stability [28]. In this experiment, the contents of Pro and soluble sugar increased under stress, and the changes in drought-resistant soybean were lower than those in sensitive soybean, which may be due to the varieties—drought-resistant soybean was less affected by drought. Abdi et al. [29] and others treated grapevine under drought stress. Their results showed that drought stress increased the contents of proline and soluble sugar in the two grapevines and reached the maximum level under severe drought, which was similar to the results of this experiment. Rudack et al. [30] conducted drought stress treatment on potatoes and found that the soluble sugar content in potatoes reached the maximum under moderate stress, and the concentration of soluble sugar did not change under more severe drought conditions. However, these results differed from ours, possibly because of the different varieties, resulting in a certain difference in the soluble sugar content in response to drought. Therefore, the stronger drought resistance of HN44 than HN65 was related to the negligible increases in MDA, H_2_O_2_, and $O_{2}^{\cdot -}$ during drought. At the same time, there were certain differences in the contributions of key regulatory substances in the two systems of antioxidant and osmotic regulation between different varieties, and the differences in contribution may be an important reason for the stronger drought resistance of HN44 than that of HN65.

## 4. Materials and Methods

### 4.1. Test Design and Materials

Soybean varieties: Heinong 44 (HN44, drought-tolerant) and Heinong 65 (HN65, drought-sensitive). The varieties were obtained from the soybean research of the Heilongjiang Academy of Agricultural Sciences (Harbin, China).

Main test instruments: soil moisture analyzer ECH2O-TE/EC-TM (EM-50, Decagon, Pullman, WA, USA), UV-Vis spectrophotometer (U-2910).

The sand culture method was used in the experiment. The diameter and height of the bucket were 0.28 m and 0.33 m, respectively. Four holes with a diameter of 1 cm were drilled into the bottom of the bucket. The bottom of the bucket was laid with gauze, and 18 kg of clean sand washed with distilled water was loaded. Eight soybean seeds were sown. Interseedlings were obtained when the first symmetrical leaf expanded. Three seedlings were preserved in each pot. The nutrient solution (500 mL) was poured once a day when the opposite leaves of the soybean were fully expanded. A nutrient solution containing 0%, 5%, 10%, 20%, and 25% PEG-6000 was poured twice a day (morning and evening) at the initial flowering stage (when a flower opened at any position on the main stem) of soybean for drought simulation treatment, 500 mL each time. The weak drought treatment used 5% and 10% PEG-6000 treatments (corresponding water potentials of −0.10 and −0.20 MPa), and the strong drought treatment used 20% and 25% PEG-6000 treatments (corresponding water potentials of −0.60 and −0.86 MPa). The treatment time was 10 days. The nutrient solution concentration is shown in Table 3. The sampling time was 8:00–9:00 a.m. The top 2 and top 3 leaves were mixed, and each treatment was repeated five times. The leaves were then stored in a refrigerator at −80 °C.

### 4.2. Determination of Membrane Lipid Peroxidation

Using the thiobarbituric acid method for the determination of the MDA content, the specific methods followed Li Hesheng “Plant physiological and biochemical experimental principles and techniques” [31]. The content of $O_{2}^{\cdot -}$ was determined by the hydroxylamine oxidation method according to the Gulcin test [32]. The content of H_2_O_2_ was determined by visible spectrophotometry according to the method of Haida et al. [33].

### 4.3. Determination of Leaf Relative Water Content and Plant Height

The relative water content (RWC) of leaves was determined by the BADR method [34], where FM, TM, and DM are the fresh, turgid, and dry masses, respectively. Three leaf discs for each accession plant exposed to drought and the corresponding control plants were cut and immediately weighed (FM), then saturated to turgidity by immersing in cold water overnight, briefly dried, weighed (TM), oven-dried at 80 °C for 24 h, and weighed again (DM).
(1)$$\mathrm{RWC}\left(100\%\right)=\frac{\mathrm{FM}-\mathrm{DM}}{\mathrm{TM}-\mathrm{DM}}\times 100$$

Soybean plant height determination reference “Soybean germplasm description specification and data standard” [35].

### 4.4. Determination of Photosynthetic Parameters of Soybean Leaves

Sunny and windless weather was selected and measured from 11:00 to 13:00 on the sampling day, The SPAD and NPQ values of soybean leaves (top three leaves) were measured using a multi-functional plant measurement instrument, MultispeQV2.0 (Beijing Huinuored Technology Co., Ltd., Beijing, China). Each treatment was repeated ten times, taking the average value. The determination temperature was 25–30 °C and the light intensity was greater than 1000 μmol·m^−2^·s^−1^.

### 4.5. Determination of Antioxidant Enzyme Activity

The nitroblue tetrazolium (NBT) method was used for the determination of SOD activity; the specific methods followed Li Hesheng “Plant physiological and biochemical experimental principles and techniques” [31]. The POD activity was determined by the guaiacol method with reference to Zhang [36]. The determination of the total antioxidant capacity (T-AOC) was performed using the corresponding commercial kit (SuzhouKeming Biotechnology Co., Ltd., Suzhou, China) according to the instructions of the manufacturer.

### 4.6. Determination of Osmotic Adjustment Substance Content

The content of Pro was determined by the sulfosalicylic acid method, and the specific method followed Zhang [36]. The soluble protein content was determined by the Coomassie brilliant blue G-250 method [37]. The soluble sugar content was determined by the anthrone method [27].

### 4.7. Contribution Analysis

Contribution of SOD, POD, and CAT to the antioxidant system:(2)$$\mathrm{SOD}\;\mathrm{Contribution}\;\left(\%\right)=\frac{\Delta \mathrm{SOD}/\mathrm{SOD}}{\left(\Delta \mathrm{SOD}/\mathrm{SOD})+(\Delta \mathrm{POD}/\mathrm{POD})+(\Delta \mathrm{CAT}/\mathrm{CAT}\right)}\;$$
where ΔSOD is the SOD activity under different water potentials minus the SOD activity in the control group, and the POD and CAT contributions to the antioxidant system (%) were calculated following the same method.

The Pro, SSC, and SP contributions to the osmotic adjustment system were determined as:(3)$$\mathrm{Pro}\;\mathrm{Contribution}\;\left(\%\right)=\frac{\Delta \mathrm{Pro}/\mathrm{Pro}}{\left(\Delta \mathrm{Pro}/\mathrm{Pro})+(\Delta \mathrm{SSC}/\mathrm{SSC})+(\Delta \mathrm{SP}/\mathrm{SP}\right)}\;$$
where ΔPro is the Pro activity under different water potentials minus the Pro activity in the control group, and the contributions (%) of SSC and SP to the osmotic adjustment system were calculated by the same method.

### 4.8. Statistical Analysis of Data

The experimental data were analyzed by IBM SPSS for the single-factor variance test, and Microsoft Office Excel 2014 (Redmond, WA, USA) and Origin Pro2021 (Origin Lab Corp., Northampton, MA, USA) for drawing.

## 5. Conclusions

This study aimed to reveal the effects of weak drought and strong drought on membrane lipid peroxidation, antioxidant enzyme activity, and osmotic adjustment substances in soybean. Soybean showed different drought characteristics under different drought conditions. Drought caused the aggravation of membrane lipid peroxidation and an increase in ROS. Plants can alleviate the negative impact of drought by increasing the levels of antioxidant enzymes and accumulating osmotic adjustment substances. Drought reduced the plant height and inhibited photosynthesis. Under weak drought, plants could resist the loss caused by drought to a certain extent, and the extension of strong drought and drought duration could cause irreversible damage to plants and even plant death. regarding the scavenging of ROS and accumulation of osmotic substances, the ability of drought-resistant variety HN44 was stronger than that of sensitive variety HN65. There were some differences in the contributions of different antioxidant enzymes and osmotic substances to drought regulation. CAT had the highest contribution in the antioxidant system, and SSC had the highest contribution in the osmotic adjustment system.

## Figures and Tables

**Figure 1 plants-11-02708-f001:**
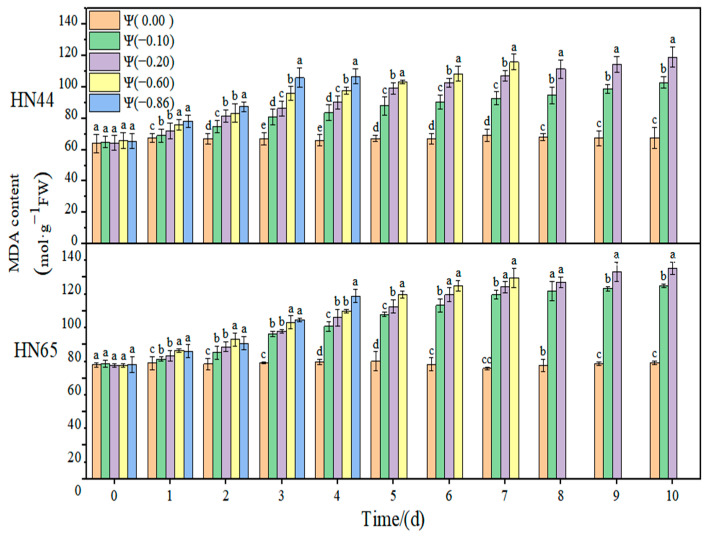
Effects of drought stress on MDA in two soybean varieties. Different letters in treatments indicate significant differences according to Duncan’s single factor variance test at the 5% level, and data are presented as mean ± SE (standard error) (n = 2).

**Figure 2 plants-11-02708-f002:**
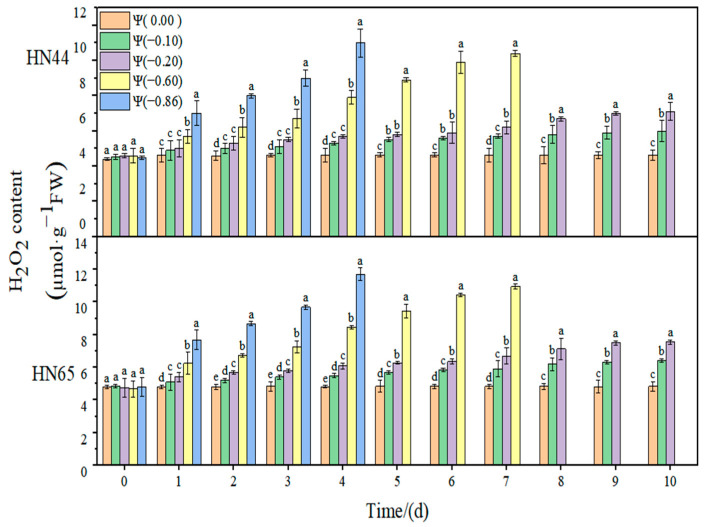
Effects of drought stress on H_2_O_2_ in two soybean varieties. Different letters in treatments indicate significant differences according to Duncan’s single factor variance test at the 5% level, and data are presented as mean ± SE (standard error) (n = 2).

**Figure 3 plants-11-02708-f003:**
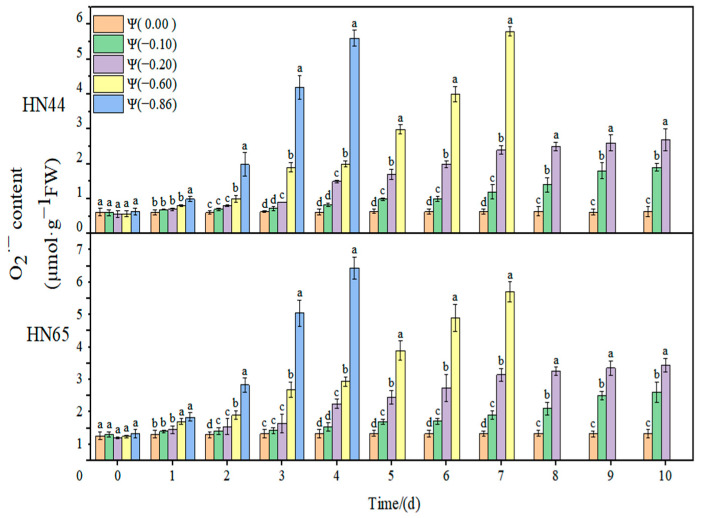
Effects of drought stress on $O_{2}^{\cdot -}$ in two soybean varieties. Different letters in treat ments indicate significant differences according to Duncan’s single factor variance test at the 5% level, and data are presented as mean ± SE (standard error) (n = 2).

**Figure 4 plants-11-02708-f004:**
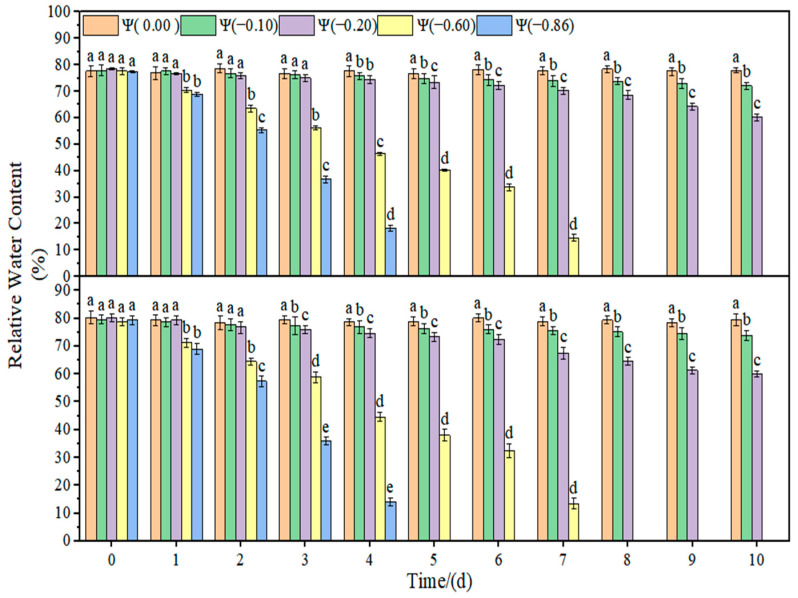
Effect of drought stress on the leaf relative water content of the two soybean varieties. Different letters in treat ments indicate significant differences according to Duncan’s single factor variance test at the 5% level, and data are presented as mean ± SE (standard error) (n = 2).

**Figure 5 plants-11-02708-f005:**
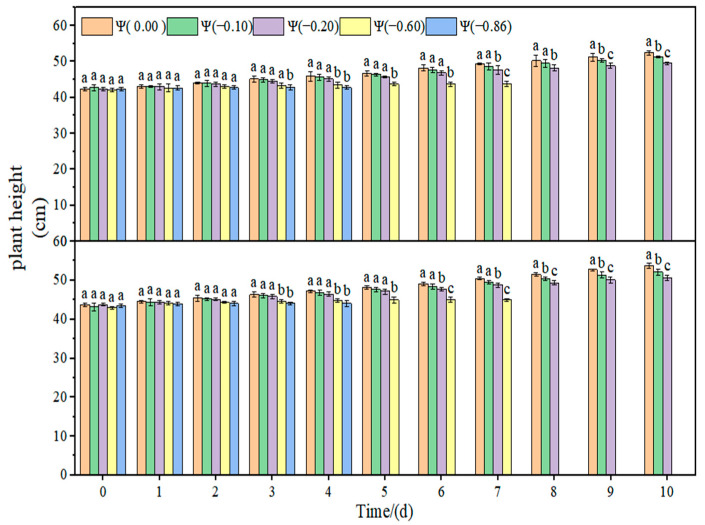
Effects of drought stress on the plant height of the two soybean varieties. Different letters in treat ments indicate significant differences according to Duncan’s single factor variance test at the 5% level, and data are presented as mean ± SE (standard error) (n = 2).

**Figure 6 plants-11-02708-f006:**
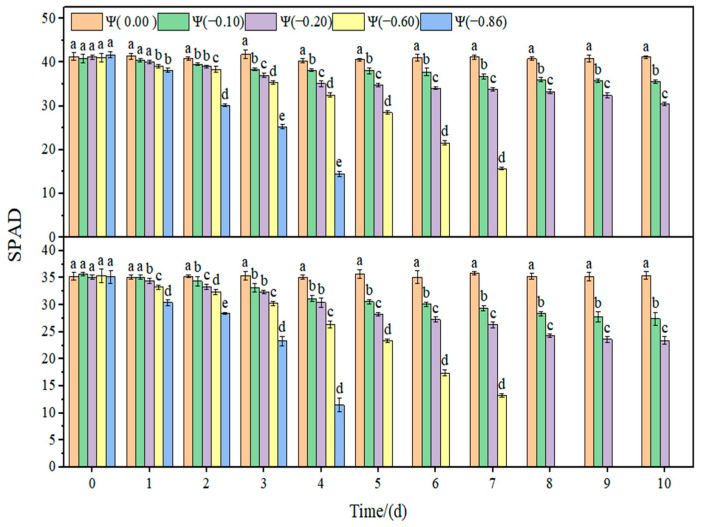
Effects of drought stress on the SPAD of the two soybean varieties. Different letters in treat ments indicate significant differences according to Duncan’s single factor variance test at the 5% level, and data are presented as mean ± SE (standard error) (n = 2).

**Figure 7 plants-11-02708-f007:**
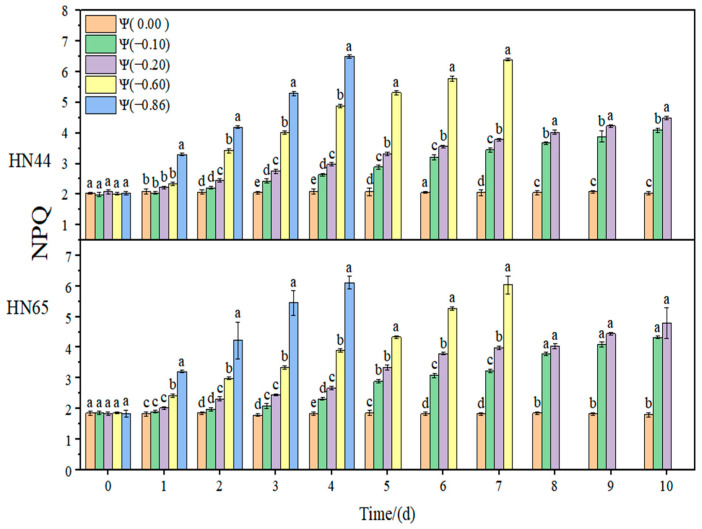
Effects of drought stress on NPQ in the two soybean varieties. Different letters in treat ments indicate significant differences according to Duncan’s single factor variance test at the 5% level, and data are presented as mean ± SE (standard error) (n = 2).

**Figure 8 plants-11-02708-f008:**
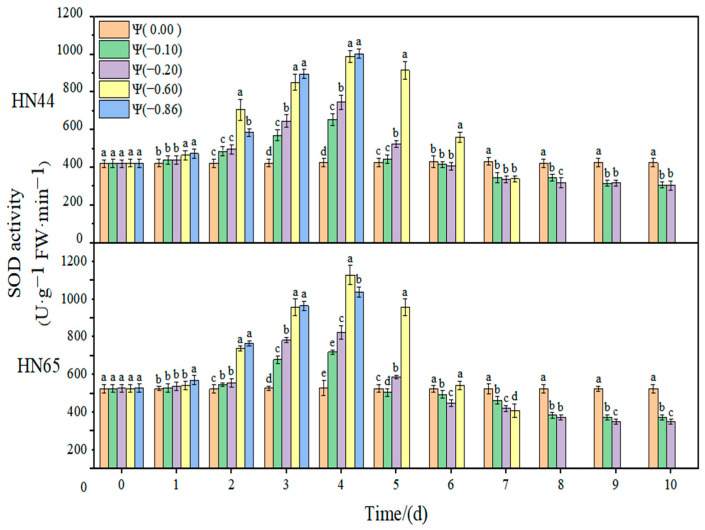
Effects of drought stress on SOD activity in the two soybean varieties. Different letters in treat ments indicate significant differences according to Duncan’s single factor variance test at the 5% level, and data are presented as mean ± SE (standard error) (n = 2).

**Figure 9 plants-11-02708-f009:**
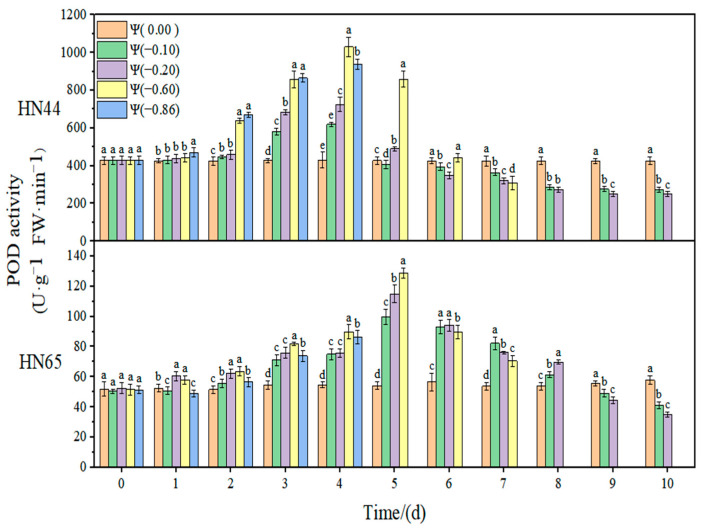
Effects of drought stress on POD activity in the two soybean varieties. Different letters in treat ments indicate significant differences according to Duncan’s single factor variance test at the 5% level, and data are presented as mean ± SE (standard error) (n = 2).

**Figure 10 plants-11-02708-f010:**
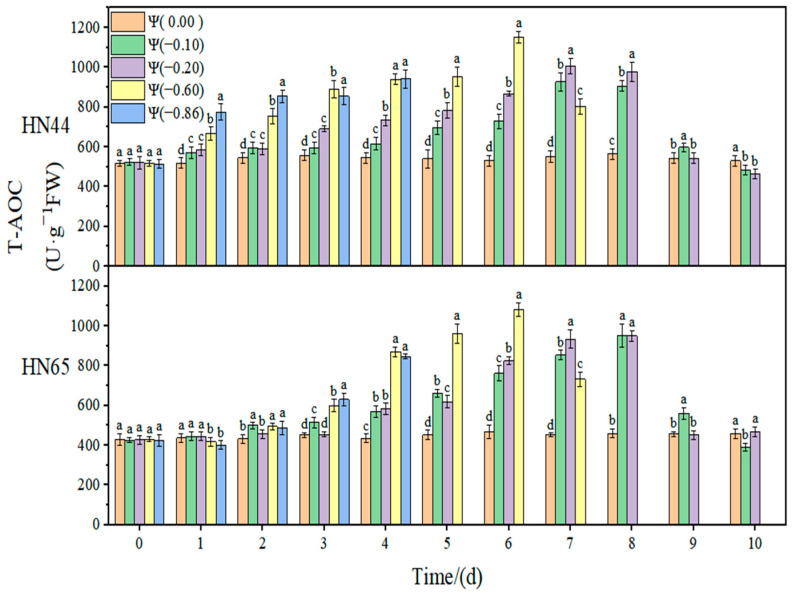
Effects of drought stress on T-AOC activity in the two soybean varieties. Different letters in treat ments indicate significant differences according to Duncan’s single factor variance test at the 5% level, and data are presented as mean ± SE (standard error) (n = 2).

**Figure 11 plants-11-02708-f011:**
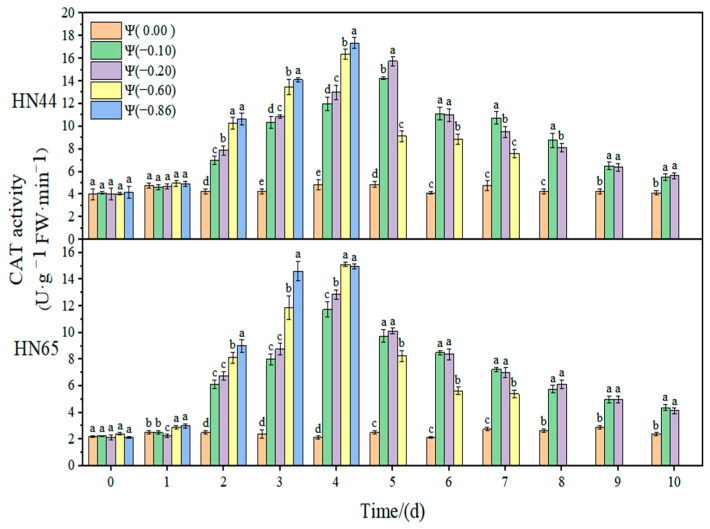
Effects of drought stress on CAT activity in the two soybean varieties. Different letters in treat ments indicate significant differences according to Duncan’s single factor variance test at the 5% level, and data are presented as mean ± SE (standard error) (n = 2).

**Figure 12 plants-11-02708-f012:**
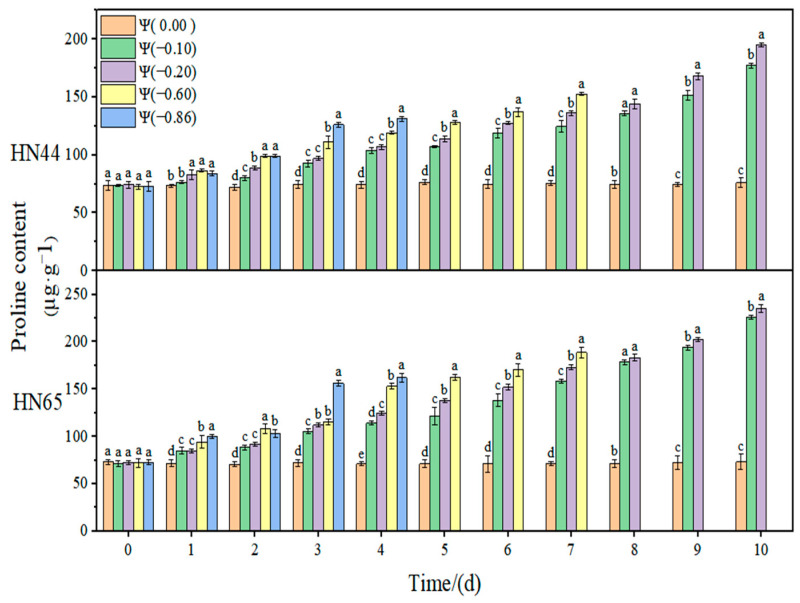
Effect of drought stress on the Pro content of the two soybean varieties. Different letters in treat ments indicate significant differences according to Duncan’s single factor variance test at the 5% level, and data are presented as mean ± SE (standard error) (n = 2).

**Figure 13 plants-11-02708-f013:**
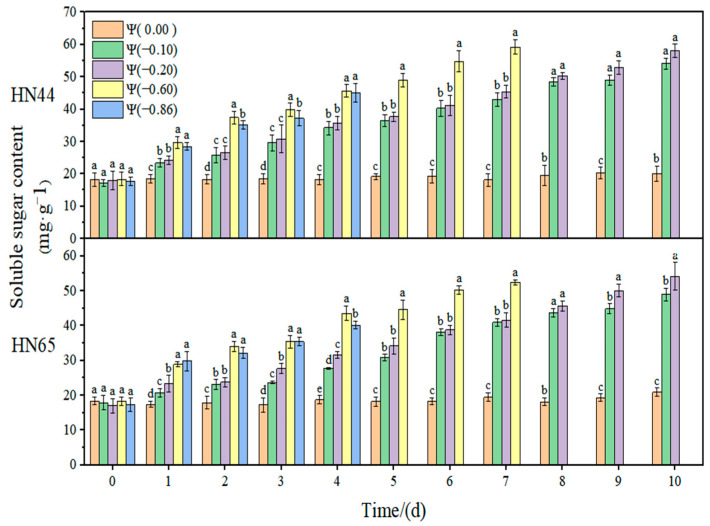
Effect of drought stress on the SSC content of the two soybean varieties. Different letters in treat ments indicate significant differences according to Duncan’s single factor variance test at the 5% level, and data are presented as mean ± SE (standard error) (n = 2).

**Figure 14 plants-11-02708-f014:**
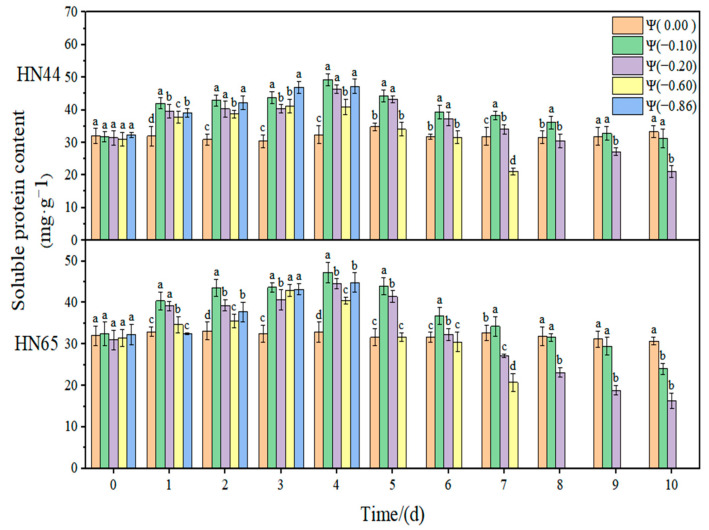
Effect of drought stress on the SP content of the two soybean varieties. Different letters in treat ments indicate significant differences according to Duncan’s single factor variance test at the 5% level, and data are presented as mean ± SE (standard error) (n = 2).

**Figure 15 plants-11-02708-f015:**
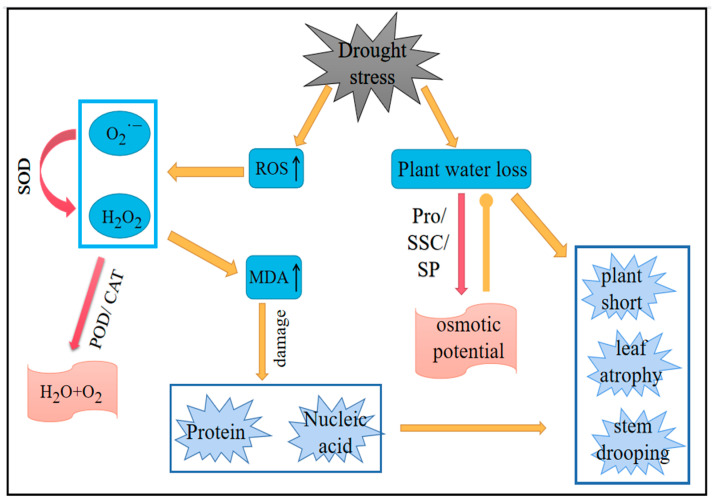
Plant damage and recovery process under drought stress. SOD, superoxide dismutase; $O_{2}^{\cdot -}$, superoxide anion; H_2_O_2_, hydrogen peroxide; POD, peroxidase; CAT, catalase; MDA, malonaldehyde; Pro, proline; SSC, soluble sugar content; SP, soluble protein. The orange arrow indicates that the damaging effect is enhanced. The pink arrow indicates that the recovery effect is enhanced and the orange circle indicates the inhibition.

**Table 1 plants-11-02708-t001:** Contribution of different antioxidant enzymes to drought regulation.

Varieties	Water-Potential (MPa)	SOD	POD	CAT
HN44	−0.10	8.80%	13.04%	78.16%
	−0.20	10.30%	16.28%	73.42%
	−0.60	15.07%	11.11%	73.82%
	−0.86	15.73%	10.08%	74.19%
HN65	−0.10	7.88%	8.57%	83.55%
	−0.20	10.63%	7.73%	81.64%
	−0.60	16.19%	9.11%	74.70%
	−0.86	14.34%	8.61%	77.05%

The results show: (1) The proportion (SOD + POD + CAT = 100%) of the three antioxidant enzymes in the antioxidant system under the same drought conditions. (2) The proportions of the three enzymes changed under different drought conditions, indicating that the three antioxidant enzymes had different contributions to drought regulation.

**Table 2 plants-11-02708-t002:** Contribution of different osmotic adjustment substances to drought resistance.

Varieties	Water Potential (MPa)	Pro	SSC	SP
HN44	−0.10	21.56%	45.81%	32.63%
	−0.20	23.46%	49.22%	27.32%
	−0.60	24.92%	60.74%	14.34%
	−0.86	27.78%	52.23%	19.99%
HN65	−0.10	34.36%	34.36%	31.28%
	−0.20	36.52%	40.60%	22.89%
	−0.60	38.22%	50.42%	11.36%
	−0.86	41.42%	43.00%	15.58%

The results show: (1) The proportions (Pro + SSC + SP = 100%) of the three osmotic adjustment substances in the osmotic adjustment system under the same drought conditions. (2) The proportions of the three regulating substances changed under different drought conditions, indicating that the three regulating substances had different contributions to drought regulation.

**Table 3 plants-11-02708-t003:** Nutrient composition.

Inorganic Salts	Concentration (mg·L^−1^)	Inorganic Salts	Concentration (mg·L^−1^)
KH_2_PO_4_	136.00	ZnSO_4_·7H_2_O	0.22
MgSO_4_	240.00	MnCl_2_·4H_2_O	4.90
CaCl_2_	220.00	H_3_BO_3_	2.86
NaMoO_4_·H_2_O	0.03	(NH_4_)_2_SO_4_	235.80
CuSO_4_·5H_2_O	0.08	Fe-EDTA	*

* Fe-EDTA: 5.57 g FeSO_4_ · H_2_O and 7.45 g Na EDTA dissolved to 1 L, respectively, with 1 mL of stock solution added per liter of nutrient solution.

## Data Availability

Not applicable.

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
