# Peer review of "Effects of Weak and Strong Drought Conditions on Physiological Stability of Flowering Soybean"

_plants, 2022, doi:10.3390/plants11202708_

Round 1
Reviewer 1 Report
Dear authors
The article on title “Effects of weak and strong drought conditions on physiological stability of flowering soybean” ” Is really interesting and describe the effects of extreme drought, weak and strong on drought-tolerant soybean Heinong 44 (HN44) and sensitive soybean Heinong 65 (HN65) using diverse physiological parameters. The manuscript is well performed and easy to follow. I think that this adds important results to consider during soybean cultivation. I recommend for acceptance with minor comments addressed below:
1.- In methodology describe the soybean varieties used in the work. Please add some information or references where it were obtained.
2.- The introduction provide sufficient background and include relevant references. But, I suggest at end of the third paragraph some references.
3.- The research design is appropriate.
4.- The methodology is adequately described.
5.- The results are clearly presented.
6.- The conclusions are supported by the results.
Author Response
The article on title “Effects of weak and strong drought conditions on physiological stability of flowering soybean” Is really interesting and describe the effects of extreme drought, weak and strong on drought-tolerant soybean Heinong 44 (HN44) and sensitive soybean Heinong 65 (HN65) using diverse physiological parameters. The manuscript is well performed and easy to follow. I think that this adds important results to consider during soybean cultivation. I recommend for acceptance with minor comments addressed below.
Response:Thank you very much for your comments, which is of great significance for us to revise the manuscript. We are glad that you like our work and have revised the manuscript according to your suggestions.
Point 1: In methodology describe the soybean varieties used in the work. Please add some information or references where it were obtained.
Response 1: Thank you for your comments, we have added the source of soybean varieties in the materials and methods section, we use soybeans from Heilongjiang Academy of Agricultural Sciences.
Point 2: The introduction provide sufficient background and include relevant references. But, I suggest at end of the third paragraph some references.
Response 2: Thank you for your comments, we added more content in the third paragraph. Contents include: Sources and functions of reactive oxygen species under normal conditions (Schneider et al., 2020; Chen et al. 2020), the harm of excessive accumulation of reactive oxygen species to plants under drought conditions (Zhang et al. 2018), and finally cited experiments to prove the relationship between reactive oxygen species and drought stress (Yang et al. 2015).
Reviewer 2 Report
Manuscript should be considered for publication after major revision in light of below comments.
1. The title of the manuscript did not justify the experimental analysis done during study. No data related to physiological parameters such as plant height and leaf expansion was measured.
2. The important parameters linked with plant response to drought stress are missing such as stomatal conductance and epidermal water loss and gas exchange.
3. In first sentence of abstract and introduction section authors mention soybean as important crop of China whereas soybean a globally important food crop. Rephrase the sentences and to strengthen the introduction section about soybean importance refer to article
“Soybean LEAFY COTYLEDON 2 regulates subsets of genes involved in controlling the biosynthesis and catabolism of seed storage substances and seed development”
4. Remove the letter ‘s’ from soybean in the list of Keywords
5. Improve introduction related to studies about soybean stress resistance by citing following article “Role of Glycine max ABSCISIC ACID INSENSITIVE 3 (GmABI3) in lipid biosynthesis and stress tolerance in soybean”.
6. At most of the places in the manuscript authors mention the term “weak and strong”, at many places the full meaning is not clear. Authors should mention “weak and strong drought stress or condition” to clarify the meaning.
7. Provide a reference for definition of early flowering stage
8. Replace word “blank group” with “control group” which is more appropriate in scientific writing
9. Restructure the first sentence of section 2.4, 4.2, 4.3 and 4.4
10. Use MS word command “Insert equation” to write and number equations in section 4.5
11. Extensive English editing, sentence structure checking and grammatical errors check is recommended
Author Response
请参阅附件。

Reviewer 3 Report
Review
for the manuscript (ID: plants-1951200) “Effects of weak and strong drought conditions on physiological stability of flowering soybean”written by Shuang Song et al.
The MS presents new data on the drought influence on the drought-tolerant soybean Heinong 44 (HN44) and sensitive soybean Heinong 65 (HN65) on such traits as the content of malondialdehyde (lipid peroxidation marker), H2O2, and O2×- (ROS), as well as the activity of superoxide dismutase, peroxidase, and catalase (antioxidant enzymes) and also the content of proline (besides an excellent osmolyte, it plays another major roles during stress), and soluble sugars (highly sensitive to environmental stress). The topic of the MS is relevant, but the content of the MS is more in line with the topics of journals on agricultural sciences and will be more interesting for readers of these journals. I recommend sending MS to such a journal.
Author Response
Thank you very much for your satisfaction with the contents of the manuscript. We received an editorial invitation for a special issue in the Plants journal “Physiological and Genetic Mechanisms of Abiotic Stress Tolerance in Crops”. The content and subject matter of the manuscript also fit well with the requirements of the special issue; at the reader level, the plants journal also includes researchers working in agriculture, botany, and plant science, and has a very broad audience. And the transfer to other journals requires additional review process, which is unfavorable for the rapid publication of articles. Therefore, we hope to continue our publishing process in this special issue, and hope that you can agree to the publication of this article in this journal.
Round 2
Reviewer 2 Report
Accept
Reviewer 3 Report
The manuscript is recommended to publish